# Low Dielectric Constant Characteristics of Styrene and Maleimide Anhydride Copolymer with Modification for High Frequency Application of Printed Circuit Board

**DOI:** 10.3390/polym15092078

**Published:** 2023-04-27

**Authors:** Pilwoo Lee, Hunsang Jung, Chan-Sei Yoo, Hyun Ho Lee

**Affiliations:** 1Department of Chemical Engineering, Myongji University, Yongin-si 17058, Gyeonggi-do, Republic of Korea; 2Electronic Convergence Materials & Devices Research Center, Korea Electronics Technology Institute, Seongnam-si 13509, Gyeonggi-do, Republic of Korea

**Keywords:** styrene maleimide copolymer, stereo-structure, dielectric constant, dissipation factor

## Abstract

Recently, due to the intensive and fast progress of the high frequency wireless communication environment, including 5th generation (5G) wireless communication, more robust substrate for printed circuit board (PCB) application, especially with less power consumption, is required. In this study, modified resins based on styrene-maleic anhydride (SMA) copolymer were prepared and evaluated as binder resin to accomplish a low dielectric constant or relative permittivity (ε_r_: <3.0) substrate for the PCB application under ultrahigh frequencies (UHF; 1 GHz~9.4 GHz). The low ε_r_ dielectric characteristics of the modified SMA copolymer could be correlated with effects from the stereo-structure of carbon chains or conformational orientation, where the degree of crystallization was analyzed by X-ray diffraction (XRD) and nuclear magnetic resonance (NMR) spectroscopies. Prepreg films of the low ε_r_ modified SMA copolymers and their compounds with epoxy resins were also characterized in terms of dielectric loss or dissipation factor (D_f_), which have shown more noticeable relation with their stereo-structures as well.

## 1. Introduction

Due to the recent ultrahigh frequency (UHF) wireless environment (fifth generation; 5G) and high integration demand for printed circuit board (PCB), the requirements for a fine line width, a large number of chip-embodiments, and a thin sheet of the PCB have been seriously pursued further than ever [1,2,3]. In other words, complicated fabrication processes have required more accommodations of large amounts of electronic elements and the high-speed operation of a device (>2 GHz) by switching acceleration simultaneously. However, to accomplish low-power consuming integrated circuits (ICs), both the low resistance of metal electrodes and the low dielectric (low ε_r_) materials of insulating layers are essential to fabricate the high-density PCB having a small operation delay (τ) of the microelectronic system [4,5,6]. Here, the τ is frequently given as a multiplicity between PCBs in the metal interconnect’s resistance (R) and capacitance (C). However, even though there have been numerous developments for the interconnects, limited attention has been given to the development of low ε_r_ materials for the PCB application.

Styrene has been widely used as a basic monomer for the microelectronic component’s layer, including the resistive switching device [7], and PCBs in various copolymer formats [8,9,10]. Meanwhile, styrene maleimide anhydride (SMA) copolymer, particularly when low molecular weighted SMA was adopted, has been widely recognized as a polymer blend compatibilizer, adhesion promoter, and binder to versatile applications including paper and microelectronics [11,12,13,14,15,16]. The paper application may contain a non-sticky coating on paper surfaces [11,16]. The microelectronic applications have been resorted to PCB formations in most cases [12,13,14,15]. Those versatile applications were enabled due to SMA’s low toxicity, good heat resistance, and dimensional stability [11,12]. Conventionally, free radical polymerization has been adopted to synthesize the SMA copolymer with polydispersities M_w_/M_n_ ~ 2 [12]. Here, numerous polymerization methods have been developed to obtain precisely or almost alternating styrene (ST) and maleimide (MA) copolymer so far. The spontaneous arrangement between nonpolar styrene and polar maleic anhydride into alternating sequences has been widely studied as a result of electron donors with the ST and acceptor with the MA complex formation at low temperature conditions (60~80 °C) [14].

However, the perfectly alternating copolymer has not been simply synthesized nor desperately required. Therefore, random distributions with monomer sequence distribution (MSD) have been preferentially utilized due to gelation continuously occurring during the copolymerization process. Separately, the SMA copolymers have been frequently analyzed by several methods, including nuclear magnetic resonance (NMR) spectroscopy. Particularly, ^13^C NMR has been often adopted to elucidate stereo-structure among ST and MA addition sequences [12,13,14,15].

The very strong tendency to form alternating ST and MA or bismaleimide could have been successfully deviated by several methods, such as controlled radical polymerization (CRP), including the introduction of nitroxide, and reversible addition-fragmentation transfer (RAFT) polymerization [17,18]. Frequently, the alternation of ST and MA during copolymerization from its electron donor-acceptor complex should be avoided to enhance the miscibility of the SMA as a compatibilizer or binder for the copper clad layer (CCL) of the PCB [19,20,21,22,23].

Epoxy resin, which is known to be a basic mixing material for the PCB, has good heat resistance, excellent insulation property, strong chemical resistance, and robust mechanical properties [24,25,26,27]. However, again, due to the rapid development of high frequency applications targeted for wireless communication (e.g., >5 GHz) technology, the limits of solely epoxy material-based PCB in terms of adaptability have been inevitably recognized. For example, a conventional epoxy resin circuit board (FR-4) under the CCL application generally has a relatively high ε_r_ (>3.0) and high dielectric loss. Therefore, the epoxy alone has insufficient susceptibility toward high frequency wireless applications [28,29].

Frequently, a high ε_r_ refers to a poor insulating property with a leakage current [25,26]. In addition, a high D_f_ refers to the energy losses that could be changed into the internal layer in the forms of vibrational or thermal energy when the voltage is applied to the nonconductive material. To diminish the energy losses from the sole epoxy material, other resins could have been mixed together for a long time due to improved mechanical properties and compatibility advantages, even with glass fibers. However, during the curing process with the epoxy resin, the ε_r_ of the mixture resin was eventually increased by the increase in the proportion of the epoxy, whose hydroxyl group often forms with permanent dipole moment [21].

Generally, the ε_r_ value of the CCL is difficult to lower by a simple modification of both extra resin and epoxy. However, the CCL layer could be controlled to lower the D_f_ by another introduction of a new binder. Therefore, the development of a third component material suitable for the PCB based on CCL is definitely desirable for low ε_r_ and low D_f_ in a wide frequency range and also lowering the τ and transmission loss.

In this study, SMA copolymer and acrylate or bismaleimide modified SMA copolymer were synthesized by copolymerization reaction along with the additional introduction of acrylate and bismaleimide monomer to the high frequency PCB application. To elucidate the correlation between the low ε_r_ dielectric properties and the copolymers’ stereochemistry, X-ray diffraction (XRD) and NMR analyses were used. Additionally, prepreg films of the SMA and acrylate or bismaleimide modified SMA copolymers were compounded with epoxy to be characterized in terms of ε_r_ and D_f_.

## 2. Materials and Methods

All the chemicals were purchased from Sigma/Aldrich unless specified. The syntheses of unmodified and modified SMA copolymers by acrylate and bismaleimide were performed as a form of the final molecular structure through three steps. First, a pre-polymer was prepared through the first reaction of ST as the monomer and benzyl peroxide (BPO) as the initiator at 115 °C with reflux unit, and the MA was dropwise added up to a molar ratio of 4:1 (ST:MA). The solvent was n-butanol with volume ratio 3 of solvent/monomer in the first reaction, and the final concentration of BPO was 6 wt.%. The reaction was stopped by autonomous gelation in case of the simultaneous introduction of ST and MA within 30 min or by prolonged reaction for 3 h with the dropwise introduction of MA. Here, the unmodified SMA could be obtained. Then, to synthesize acrylate or bismaleimide modified SMA, an acrylate monomer or bismaleimide monomer accompanied by the MA in n-butanol were added in a dropwise manner for 3 h into the pre-polymer before the gelation for each polymer [17,19,20,21].

Secondly, polymerization processes proceeded with a molar ratio of 4:1:0.03 (ST:MA:acrylate or bismaleimide) through an aging reaction of 3 h in which an aromatic-anhydride pre-polymer was firstly formed. Here, the products of final modified resins were in gelatin-like forms as well.

Finally, after the synthesis, the temperature was slowly reduced to 40 °C. To purify the gelatin-like product, deionized (DI) water/methanol (vol. ratio of 1:1) was added three times. Then, 30 min stirring and subsequent 30 min incubation could provide two separated phases, which can collect or extract with purified polymer portions at the bottom phases. The harvested polymer was suction-filtered and the subsequent filtered portion was dissolved in toluene. The synthesized polymer in toluene was heated to 180 °C to remove any remaining volatile solvent or chemicals.

For the measurement of the dielectric constant of each frequency, prepreg films without copper electrode layers were fabricated in a method of solution process. Each polymer sample (SMA, acrylate SMA, bismaleimide SMA) was dissolved in dimethylformamide (DMF) in a weight ratio of 10% and 20%. To prepare sample film for the dielectric property test, 1:1 molar ratio of epoxy (YD-128, Kukdo Chemical Co., Seoul, Korea) and the three synthesized SMA copolymers were mixed with the addition of 2-imidazole as a hardening enhancer. The film was baked on aluminum foil from 70 °C to 150 °C for 4 h. The film thicknesses were kept as 0.8 mm.

For dielectric properties, the Agilent N5230A microwave network analyzer (1 MHz to 10 GHz) was used to directly measure ε_r_ and D_f_ values of the SMA, and modified SMA polymer’s flat film (0.8 mm) at room temperature without any support or electrode. For the measurement, five different spots were tested and the averaged values were plotted.

The characterization of the physical properties of the unmodified SMA, acrylic SMA, and bismaleimide SMA copolymers right after their syntheses were performed to measure the number averaged molecular weight (M_n_), weight averaged molecular weight (M_w_), polydispersity index (PDI), and glass transition temperature (T_g_), respectively. The Mn, Mw, and PDI were measured by gel permeation chromatography (GPC; 1260 Infinity II, Agilent Co., Colorado Springs, CO, USA). The T_g_ was measured by differential scanning calorimeter (DSC; 131evo, Setaram KEP Technologies, Plan-les-Ouates, Switzerland) under N_2_ environment with a programmable temperature gradient of 20 °C/min.

The chemical and electrical characterizations of the unmodified SMA, acrylic SMA, and bismaleimide SMA copolymers have been compared, respectively. Through XRD and NMR (700MHz, Bruker, Billerica, MA, USA) analyses, the synthesis mechanism was proposed and the molecular structure was thoroughly investigated. In addition, monomer conformational property in the SMA copolymer was pursued by ^13^C NMR analysis rather than ^1^H NMR analysis [11,12,19].

## 3. Results

Modification methods toward additional third monomer integration other than unwanted styrene and maleimide were introduced to improve properties such as an excellent ε_r_ characteristic and D_f_ characteristic. Figure 1 shows a schematic modification method for aromatic-anhydride copolymer during polymerizing copolymerization. At the same time, dried copolymer powders or fragments after the purification procedures were shown as photographic images in Figure 1 for the acrylate and bismaleimide modified SMA copolymers. As shown in Figure 1, the modification methods to replace the anhydride group with a similar derivative material or others including acrylate and bismaleimide were devised.

Table 1 shows the number average molecular weight (M_n_), averaged molecular weight (M_w_), polydispersity index (PDI), and T_g_ of the unmodified SMA, acrylic SMA, and bismaleimide SMA copolymers, respectively. With the modification of acrylate and bismaleimide to the SMA, both M_n_ and M_w_ increased as expected, which could provide suitable thermal stability with the higher T_g_ for the high frequency applications. In addition, the bismaleimide modified SMA shows the lowest values of PDI.

Figure 2 shows XRD spectra analysis representing polymer crystallinity. Peaks around 20° and 42° are represented with the ST and the MA, respectively. Moreover, the additional peak around 10° is divided into a plurality of peaks in the case of high crystallinity. However, the broad peak means that the three synthesized SMAs represent the amorphous state.

Figure 3 shows ^13^C NMR chemical shifts of SMA, acrylate modified SMA, and bismaleimide modified SMA. It has been well known that the tacticity of copolymer can have a significant effect on the chemical shifts of ^13^C NMR spectra [11,12,19].

Table 2 shows a downfield C7 peak at an aromatic ring in styrene. Depending on the orientation of the benzene ring and imide ring, four different triads, such as SSS, MSS (SSM), and MSM, can exist. From many literatures, it has been recognized that peaks around 172 ppm correspond to carbonyl carbon (C5 or C6) on maleic anhydride, peaks around 137~147 ppm to the C7 atom of the styrene aromatic ring, peaks around 128 ppm to the C8~C12 carbons of the styrene aromatic ring, and peaks around 42~51.6 ppm to methine and methylene carbons (C1 or C2) at backbone chain, as shown in Figure 3 [11,12,19].

In this study, for the unmodified SMA, acrylate SMA, and bismaleimide SMA copolymers, three broad resonance peaks from C7 peaks marked as SSS, SSM (MSS), and MSM triads could be obtained, as shown in Figure 3 and explained in Table 2.

As shown in Figure 3 and Table 3, there are two fingerprint regions for triads of SMA copolymer. The ^13^C peaks corresponded to methylene spectra from -CH, -CH_2_, and -CH_3_. Those peaks have been identified to give triad fractions of the SMA copolymer. Those triads are illustrated in Figure 3 as well. In the very early period with ^13^C NMR spectra, special techniques including spin-echo and polarization transfer have been utilized to investigate the triads [11,19]. However, the recent ultrahigh frequency (>700 MHz) NMR instrument could enable a better resolution to induce less overlapping between peaks. As shown in Table 3, at the MSM and SSM (MSS) region (136 to 147 ppm), the intensity of 142 to 147 ppm increases when acrylate and bismaleimide were added.

Figure 4 shows the ^1^H NMR spectrum of each SMA polymer. The region 5.8~7.6 ppm indicates styrene’s aromatic ring of hydrogen and the region of 2.2~3.8 ppm methylene/methane group in polymer backbone in this 3.7~3.1 ppm region indicates maleic anhydride’s methane proton peaks [12,15], and acrylic modified SMA shows a peak near 1 ppm because of the methyl of terminal acrylic chain [17,19].

From ^1^H NMR and ^13^C NMR chemical shift spectra, mole fractions of ST and MA can be estimated by integrated peak area intensities. In ^1^H NMR spectra, the intensity of phenyl proton (*I*_phenyl_) and total proton (*I*_total_) peaks according to Equations (1) and (2). According to this method, the relative error range was ±2%. The calculation results are shown in Table 4.
(1)$$mol\;fraction\;MA=\frac{5I_{total}-8I_{total}}{5I_{total}-6I_{total}}$$
(2)$$mol\;fraction\;ST=\frac{2I_{phenyl}}{5I_{total}-6I_{total}}$$
(3)$$\frac{mol\;fraction\;MA}{mol\;fraction\;ST}=\frac{(I_{170\sim 175\;ppm})/2}{(I_{133\sim 150\;ppm}+I_{C_{st}(120\sim 130\;ppm)})/6}$$
(4)mol fraction MA+mol fraction ST=1
with ^13^C NMR spectra, Equations (3) and (4) have the advantage that they are not affected by the presence of water, which is often difficult to completely remove due to the hydrophilicity of MA.

Figure 5 shows ε_r_ and D_f_ data of reference, pristine SMA, and acrylic modified and bismaleimide modified SMA. Each SMA was measured from UHF environments to super high frequency (SHF; 3~30 GHz) conditions ranging from 1 GHz to 9.4 GHz.

As shown in Figure 5, the bismaleimide modified SMA shows the lowest dielectric constants ranging from 1.4 to 1.8, and the lowest dissipation factor under 0.03 as well, even under the ultrahigh frequency of 9.4 GHz. The results matched the low PDI values of the bismaleimide modified SMA, as shown in Table 1.

## 4. Discussion

The styrene-anhydride copolymer is a robust polymer, which has both strictly hydrophobic (aromatic monomer) and less hydrophobic (maleic anhydride) functional moiety. It shows various properties such as hydrophobic interactions, hydrogen bonding, and ionic action with the presence of polar and non-polar functionalities in the copolymer structures [11,12,13,14]. Since the MA can be partially esterified in the random format, the partially esterified MA was believed to be one reason for random copolymer formation rather than perfectly alternating SMA. Moreover, the high solubility of the SMA copolymer with various organic solvents is another important advantage of the SMA copolymer as a PCB binder [15,16,17].

Frequently, the tacticity of the PS polymer is technologically very important for an electronic substrate application. Meanwhile, the atactic PS materials have ambiguous properties because polymer chains could not be regularly stacked due to the irregular structure. Therefore, while syndiotactic PS can be found to be a semicrystalline material, the atactic PS, which is a common case in the conformations of the PS, could not be crystallized to be in a glass phase. However, the stereo-structure can be altered and cured by the purification method or the introduction of a third monomer during copolymerization [20,21].

As illustrated in Figure 2, broad peaks around the 10–15° region of 2θ were apparently shown for all SMA copolymers, which represent the semicrystalline phases out of PS fractions [29]. Among the three SMA copolymers, the unmodified SMA copolymer shows a relatively higher peak of the semicrystalline PS than the other two modified SMAs, which can also support the unmodified SMA data of Table 1 with a low PDI value and high triad ratio of SSS, as shown in Table 3.

Moreover, SMA copolymer resins with high functionality have been prepared by adding imide or acrylic monomers to improve the thermal, adhesion, and electrical properties in aromatic monomer and anhydride structures [20,22,26]. The SMA copolymer was synthesized using aromatic and anhydride monomer by chain polymerization under a thermally working condition. The final molecular structure could be formed with a randomly mixed copolymer or partial block polymer [20,21,22,23,26].

As shown in Table 1, the polymerization properties of the acrylate or bismaleimide modified SMA copolymers exhibited a higher T_g_ than the unmodified SMA copolymer, which would induce thermal stability upon PCB application under ultrahigh frequency. In fact, to enforce the thermal stability, inorganic nanocomposites have been continuously introduced [5]. However, there could be mixing or blending difficulties with the inorganic composites [4,5]. In addition, various initiator concentrations, ST:MA feed ratios (1:1, 2:1, 4:1, 6:1), and reaction temperatures (110 °C, 115 °C, 120 °C, 125 °C) have been examined in this study. In Table 1, the reaction conditions achieving a PDI value under 4.0 were kept for all SMA copolymers [19].

Generally, during the copolymerization process, a monomer of a high reaction activity was polymerized into the copolymer first, when the remaining second monomers were added at the same time. Then, the monomer of the lowest reaction activity was bonded later to form a block or random copolymer [19,20]. Therefore, in this study, the modifiers of the acrylate and bismaleimide about the SMA were added during the copolymerization process along with the MA considering the highest reactivity of the ST.

Depending on the stereo-structure or configuration of the carbon backbone determined by the position of the benzene ring, they can divide into atactic, isotactic, and syndiotactic alignments [11,14,15]. The stereo-structure can generate a difference in the microstructural regularity of the PS. In addition, when the syndiotactic styrene was ideally synthesized using a metallocene catalyst, they were known to be superior in heat resistance, and its chemical resistance was better than atactic and isotactic, which are random structures due to its stereo-regularity [19,25].

Figure 3 shows the ^13^C NMR spectra of SMA, acrylate modified SMA, and bismaleimide modified SMA. It has been well known that the tacticity of copolymer can induce a significant effect on the chemical shifts of ^13^C NMR spectra [11,12,19]. For ^13^C NMR spectra, the chemical shift and broad splitting were originated by stereochemical configuration and copolymer sequence [15]. Conventionally, the carbons in the benzene ring have shown to have three splitted groups, which could also be found in Figure 3.

As shown in Figure 3 and Table 3, the unmodified SMA copolymer’s ST mole fraction was higher than those of the two modified SMA copolymers. In fact, without the dropwise addition of the MA during polymerization, there could be the very fast formation of alternating ST and MA copolymer in a short time. However, the simultaneous introduction of ST and MA resulted in the termination of the synthesizing process due to the extremely high viscosity. Therefore, the dropwise introduction of the MA was inevitable in this study to ensure there was sufficient time to modify the random SMA copolymer. In addition, the complete random copolymer of irregular carbon chain structure was formed by using individual monomers having a similar molecular reaction activity, respectively [19,20,21,22,23,26]. The irregular orientation of carbon chains was clearly identified in Table 3, as all ratios of SSS of the unmodified (0.791), acrylate modified (0.186), and bismaleimide (0.187) modified SMAs were significantly different from alternating SMA copolymers having the discrete electron donor (ST) and acceptor (MA) complex identified in the NMR ^13^C peak shift analysis [11,12,14].

Based on the ^1^H NMR peaks of Figure 4, the conversion of double bond can be identified for the bismaleimide modified SMA copolymer [18]. Due to the expected poor activity of the bismaleimide’s double bond compared with the acrylate’s double bond originated from steric hindrance, the conversion of the double bonds needs to be examined to postulate overall reaction efficiency. As shown in Figure 5, the strong peak around 2.85 ppm (H of methyl group attached to nitrogen atom of the bismaleimide) for the bismaleimide modified SMA was detected along with broad peaks around 5.22~6.67 ppm, which were known to correspond to the conversion of styrene’s double bonds [18].

The region of 5.8~7.6 ppm indicates styrene’s aromatic ring of hydrogen and the region of 2.2~3.8 ppm methylene/methane group in polymer backbone in the 3.7~3.1 ppm region indicates maleic anhydride’s methane proton peaks [12,15], and the acrylic modified SMA shows a peak near 1 ppm because of the methyl of terminal acrylic chain [17,19].

When epoxy resin is reacted and mixed with the synthesized SMA copolymer, the functional group of anhydride became ring opened and bound with the functional group of epoxy to form crosslinking as a hardening agent [24,25,26]. After preparations of prepreg films with the epoxy resin, the measured values of T_g_ were obtained as having increased to 165.9 °C for the unmodified SMA, 173.2 °C for the acrylate modified SMA, and 172.8 °C for the bismaleimide modified SMA, respectively. The mol fraction of ST was decreased by the acrylate and bismaleimide, as shown in Table 3. The mole fraction can be influenced by the stereoregularity, which can affect the ε_r_ and D_f_ value.

As shown in ε_r_ and D_f_ data of Figure 5a,b, pristine SMA, acrylate SMA, and bismaleimide SMA have shown distinct SHF characteristics, respectively. In Figure 5a, all ε_r_ values of SMA copolymers showed a tendency to decrease with the increasing frequency from UHF to SHF [30].

The pristine SMA could show a lower ε_r_ value under 3.0 at each frequency. However, the modified SMAs copolymers had much lower ε_r_ values than the unmodified SMA. Particularly, for the ε_r_ values under 2.0, it is found that the bismaleimide SMA copolymer has a relatively large deviation, while the acrylate SMA copolymer shows little deviation. As shown in Figure 5a and Table 4, two modified copolymers have shown a distinctly lower ε_r_ and lower mol fraction of MA in ^13^C peak than the unmodified one.

As shown in Figure 5b, all D_f_ values have a tendency to increase with the increase in frequency. In addition, all SMA copolymers show an increase at the deviation of D_f_ value as the frequency increased. In addition, two modifications of the SMA copolymer could reduce ε_r_ and D_f_ when compared with those of the pristine or unmodified one. Therefore, the modified SMA copolymers have a higher suitability for the next generation substrate under SHF wireless communication.

For the issues of low ε_r_ dielectric substrate, various synthesis methods using maleic anhydride and epoxy have been reported [29]. However, there needs to be specific formations of multiple oligomers mixed with various monomers, including polybutadiene (PB), polyimide (PI), and polytetrafluoroethylene (PTFE), which should be filmed as double-sided coatings [29]. Therefore, through this study, relatively simple modifications on SMA copolymer can be proposed for a high performance of PCB substrate.

## 5. Conclusions

In this study, acrylate and bismaleimide modified resins of SMA copolymer were evaluated as binder resin to accomplish a very low ε_r_ and D_f_ substrate for PCB application under a UHF of 1 GHz~9.4 GHz. The low ε_r_ and D_f_ dielectric characteristics of the modified SMA copolymer were found to significantly depend on the stereo-structure of carbon chains or conformational orientation. The conformations by the stereo chemistry were analyzed by XRD and NMR measurements to elucidate the correlation between dielectric properties and copolymers’ stereochemistry.

## Figures and Tables

**Figure 1 polymers-15-02078-f001:**
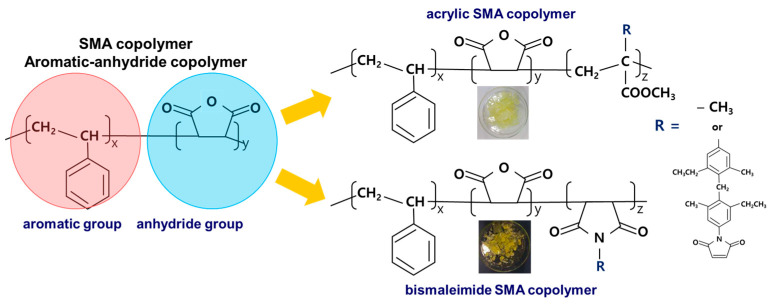
A schematic diagram of the aromatic-anhydride copolymer modification method.

**Figure 2 polymers-15-02078-f002:**
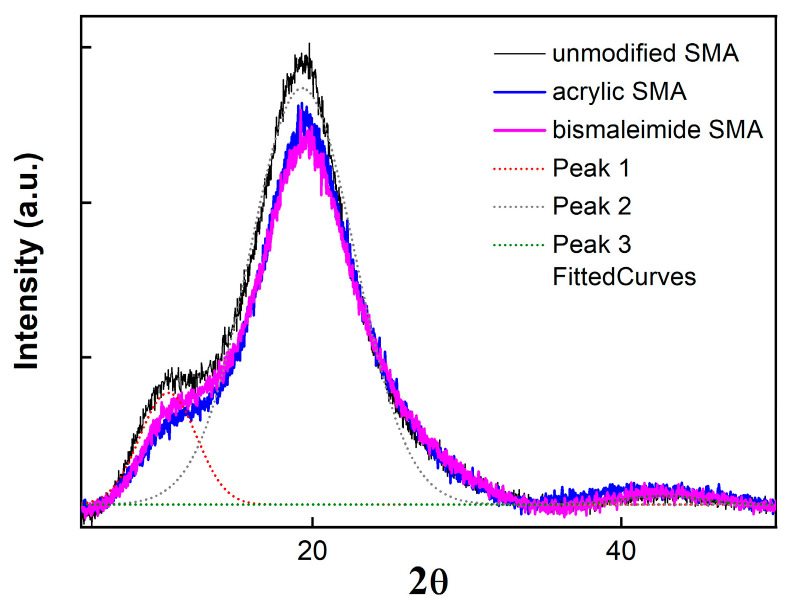
XRD spectra patterns of unmodified SMA copolymer and acrylate or bismaleimide SMA copolymer.

**Figure 3 polymers-15-02078-f003:**
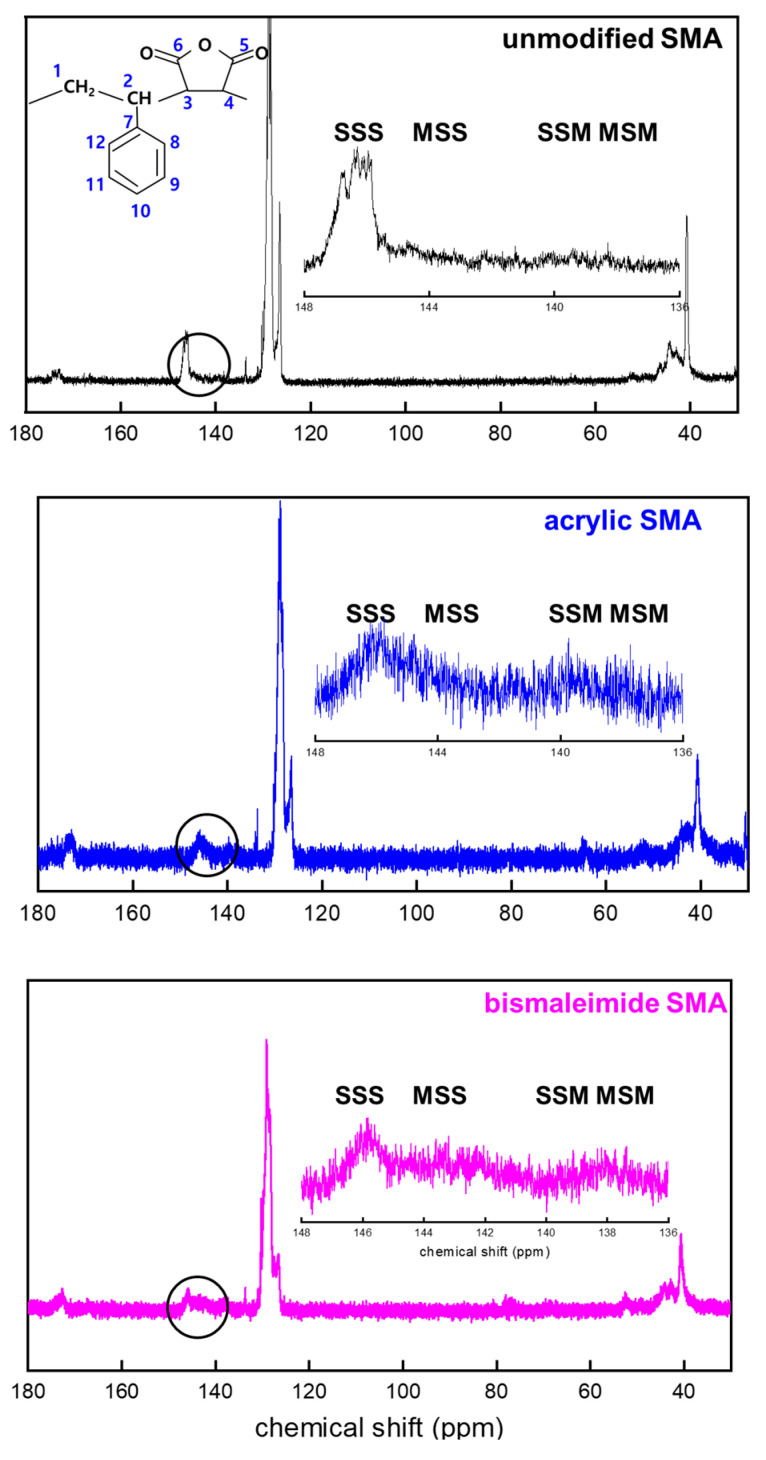
^13^C NMR chemical shift peaks of aliphatic CH and CH_2_ polymer chain of unmodified SMA copolymer, acrylate SMA copolymer, and bismaleimide SMA copolymer.

**Figure 4 polymers-15-02078-f004:**
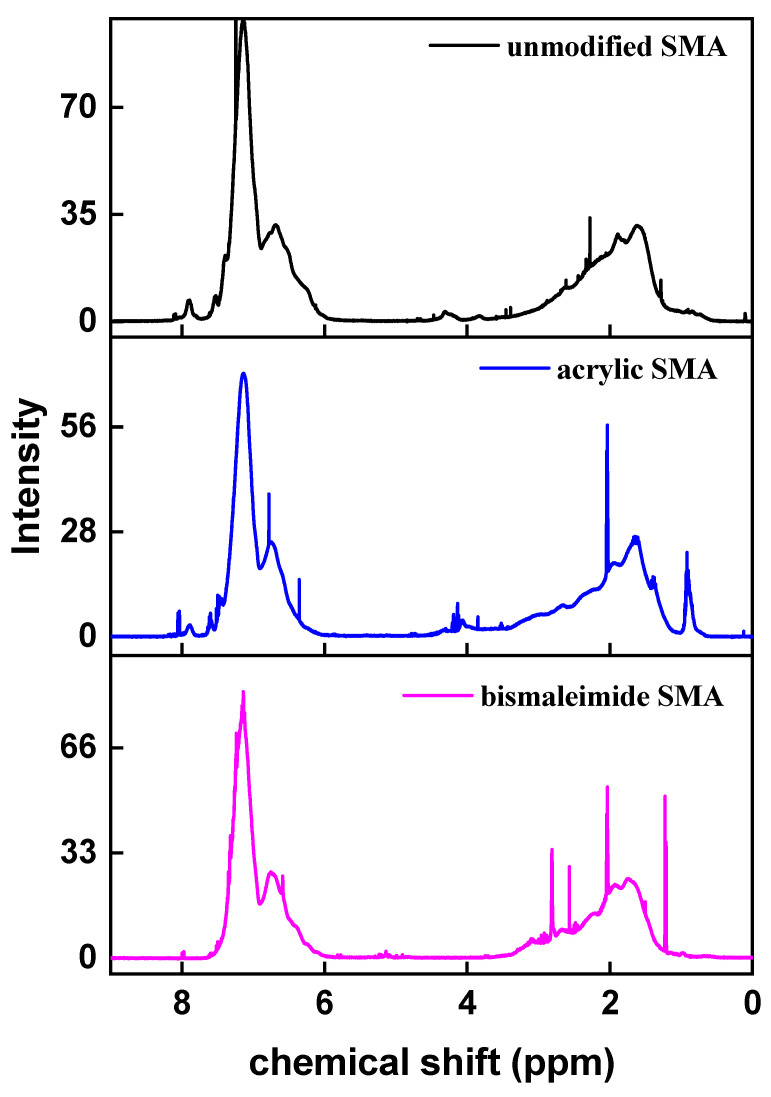
^1^H NMR chemical shift peaks of aliphatic CH and CH_2_ polymer chain of unmodified SMA copolymer, acrylate SMA copolymer, and bismaleimide SMA copolymer.

**Figure 5 polymers-15-02078-f005:**
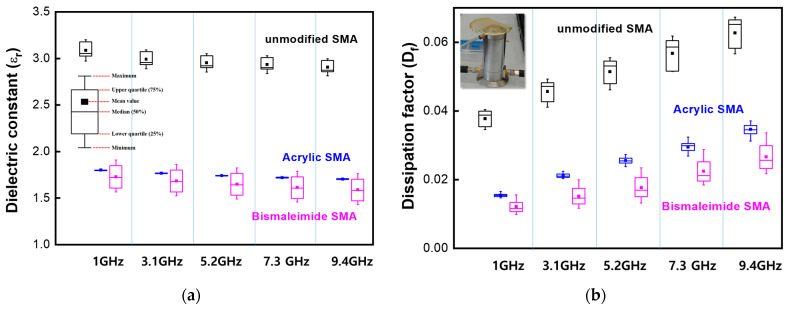
(**a**) ε_r_ and (**b**) D_f_ values measured by Agilent N5230A microwave network analyzer.

**Table 1 polymers-15-02078-t001:** Number average molecular weight (M_n_), averaged molecular weight (M_w_), polydispersity index (PDI), and T_g_ of the unmodified SMA, acrylic SMA, and bismaleimide SMA copolymers.

	M_n_	M_w_	PDI	T_g_ (°C)
unmodified SMA	3469	10,921	3.15	104.4
acrylic SMA	3982	20,103	3.75	113.0
bismaleimide SMA	5897	18,925	3.21	112.0

**Table 2 polymers-15-02078-t002:** Downfield C7 peaks at aromatic ring in styrene from ^13^C NMR chemical shift of Figure 3.

Triad	^13^C NMR Chemical Shift	Array Molecular Structure	
SSS	147~150 ppm	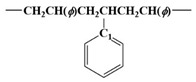	
MSS/SSM	142~147 ppm	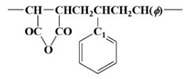	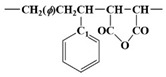
MSM	136~140 ppm	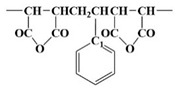	

reference [11].

**Table 3 polymers-15-02078-t003:** Monomer sequence distribution of unmodified SMA copolymer, acrylate SMA copolymer, and bismaleimide SMA copolymer from ^13^C NMR chemical shift of Figure 3.

	Triad Ratio		
SMA	SSS	SSM/MSS	MSM
unmodified SMA	0.791	0.101	0.108
acrylic SMA	0.186	0.508	0.306
bismaleimide SMA	0.187	0.490	0.323

**Table 4 polymers-15-02078-t004:** Mole fractions of ST (mol fraction ST) and MA (mol fraction MA) from ^1^H NMR and ^13^C NMR chemical shift spectra.

	mol Fraction in ^1^H NMR		mol Fraction in ^13^C NMR	
	mol Fraction MA	mol Fraction ST	mol Fraction MA	mol Fraction ST
unmodified SMA	0.112	0.888	0.188	0.812
acrylic SMA	0.439	0.561	0.261	0.739
bismaleimide SMA	0.290	0.710	0.257	0.743

## Data Availability

Not applicable.

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
