# Peer review of "Low Dielectric Constant Characteristics of Styrene and Maleimide Anhydride Copolymer with Modification for High Frequency Application of Printed Circuit Board"

_polymers, 2023, doi:10.3390/polym15092078_

Round 1
Reviewer 1 Report
In this manuscript, the author reported a low dielectric constant modified copolymer for PCB application, which is a meaningful work. However, there is a serious research mistakes, therefore, I don’t recommend the publication.
Some issues were listed as follow:
(1) The synthesis procedure for the copolymer should be present in detail, such as reaction temperature and time.
(2) The preparation procedure for the solution process should be present in detail, dried at RT or other temperature?
(3) In the section of abstract, introduction and result, there is a sample consistent of SMA and epoxy, however, there is not introduce of this sample in the section of materials and methods, why?
(4) What kind of epoxy were used?
(5) Line 251, “When epoxy resin is reacted and mixed with the synthesized SMA copolymer, the functional group of anhydride became ring opened and bound with functional group of epoxy to form crosslinking as a hardening agent” Why there was no synthesis procedure for the reaction of epoxy and SMA?
(6) Why there is no structure and properties investigation for the sample containing epoxy?
(7) What is the Mn, Mw and PDI for the different obtained polymer? Which is very important for the synthetic polymer.
(8) How is the thermal and mechanical properties for the obtained polymers?
Author Response
Reply and Response (Polymers 2281088)
*All corrected paragraphs in revised manuscript were blue-colored.
Corrected parts were highlighted in blue color, at the same time, “track changes” function was in mode during revision.
Reviewer #1 :
In this manuscript, the author reported a low dielectric constant modified copolymer for PCB application, which is a meaningful work. However, there is a serious research mistakes, therefore, I don’t recommend the publication.
Some issues were listed as follow:
(1) The synthesis procedure for the copolymer should be present in detail, such as reaction temperature and time.
Response: Authors would like to thank reviewer for valuable comments. Very detailed reaction temperature and time was given in revised manuscript.
In lines of 102-125,
“The syntheses of unmodified and modified SMA copolymers by acrylate and bismaleimide were performed as a form of the final molecular structure through three steps. First, a pre-polymer was prepared through the first reaction of ST as monomer and benzyl peroxide (BPO) as initiator at 115 oC with reflux unit and the MA was dropwise added up to molar ratio of 4:1 (ST:MA). The solvent was n-butanol with volume ratio 3 of solvent/monomer in the first reaction and the final concentration of BPO was 6 wt.%. The reaction was stopped by autonomous gelation in a case of simultaneous introduction of ST and MA within 30 min or by prolonged reaction for 3 hours with the drop-wise introduction of MA. Here, the unmodified SMA could be obtained. And then, to synthesize acrylate or bismaleimide modified SMA, an acrylate monomer or bismaleimide monomer accompanied by the MA in n-butanol were added in a dropwise man-ner for 3 hours into the pre-polymer before the gelation for each polymer [17,19,20,21].
Secondly, polymerization processes were proceeded with molar ratio of 4:1:0.03 (ST:MA:acrylate or bismaleimide) through an aging reaction of 3 hours in which an aromatic-anhydride pre-polymer was firstly formed. Here, products of final modified resins were in gelatin-like forms as well.
Finally, after the synthesis, the temperature was reduced slowly to 40 oC. Then, to purify the gelatin-like product, DI water/methanol (vol. ratio of 1:1) was added three times. 30 min stirring and subsequent 30 min incubation could provide two separated phases, which can collect or extract with purified polymer portions at the bottom phases. The harvested polymer was suction-filtered and the subsequent filtered portion was dissolved in toluene. The synthesized polymer in toluene was heated to 180 oC to remove any remained volatile solvent or chemicals.”
(2) The preparation procedure for the solution process should be present in detail, dried at RT or other temperature?
Response: Authors would like to thank reviewer for important comments. Very detailed reaction temperature and time was given in revised manuscript.
In lines of 102-125,
“The syntheses of unmodified and modified SMA copolymers by acrylate and bismaleimide were performed as a form of the final molecular structure through three steps. First, a pre-polymer was prepared through the first reaction of ST as monomer and benzyl peroxide (BPO) as initiator at 115 oC with reflux unit and the MA was dropwise added up to molar ratio of 4:1 (ST:MA). The solvent was n-butanol with volume ratio 3 of solvent/monomer in the first reaction and the final concentration of BPO was 6 wt.%. The reaction was stopped by autonomous gelation in a case of simultaneous introduction of ST and MA within 30 min or by prolonged reaction for 3 hours with the drop-wise introduction of MA. Here, the unmodified SMA could be obtained. And then, to synthesize acrylate or bismaleimide modified SMA, an acrylate monomer or bismaleimide monomer accompanied by the MA in n-butanol were added in a dropwise man-ner for 3 hours into the pre-polymer before the gelation for each polymer [17,19,20,21].
Secondly, polymerization processes were proceeded with molar ratio of 4:1:0.03 (ST:MA:acrylate or bismaleimide) through an aging reaction of 3 hours in which an aromatic-anhydride pre-polymer was firstly formed. Here, products of final modified resins were in gelatin-like forms as well.
Finally, after the synthesis, the temperature was reduced slowly to 40 oC. Then, to purify the gelatin-like product, DI water/methanol (vol. ratio of 1:1) was added three times. 30 min stirring and subsequent 30 min incubation could provide two separated phases, which can collect or extract with purified polymer portions at the bottom phases. The harvested polymer was suction-filtered and the subsequent filtered portion was dissolved in toluene. The synthesized polymer in toluene was heated to 180 oC to remove any remained volatile solvent or chemicals.”
(3) In the section of abstract, introduction and result, there is a sample consistent of SMA and epoxy, however, there is not introduction of this sample in the section of materials and methods, why?
Response: Authors thank for helpful comments. Authors added more descriptions and 3 reference paper how the SMA and epoxy sample can be made and their chemical position in revised manuscript.
In lines of 127-134,
“the dielectric constant of each frequency, prepreg films without copper electrode layers were fabricated in a method of solution process. Each polymer sample (SMA, acrylate SMA, bismaleimide SMA) was dissolved in dimethylformamide (DMF) in a weight ratio of 10 % and 20 %. To prepare sample film for the dielectric property test, 1:1 molar ratio of epoxy (YD-128, Kukdo Chemical Co., Korea) and the three synthesized SMA copolymers were mixed with addition of 2-imidazole as a hardening enhancer. The film was baked on aluminum foil from 70 oC to 150 oC for four hours. The film thicknesses were kept as 0.8 mm.”
(4) What kind of epoxy were used?
Response: Authors would like to thank reviewer for pointing out the missing information. The epoxy was a commercialized product of YD-128, Kukdo Chemical Co., Korea.
(5) Line 251, “When epoxy resin is reacted and mixed with the synthesized SMA copolymer, the functional group of anhydride became ring opened and bound with functional group of epoxy to form crosslinking as a hardening agent” Why there was no synthesis procedure for the reaction of epoxy and SMA?
Response: Authors are grateful for reviewer to point out the missing information. The procedure to make prepreg film was added in manuscript.
For the synthesis procedure, it is given as follows:
In lines of 127-134,
“For the measurement of the dielectric constant of each frequency, prepreg films without copper electrode layers were fabricated in a method of solution process. Each polymer sample (SMA, acrylate SMA, bismaleimide SMA) was dissolved in dimethylformamide (DMF) in a weight ratio of 10 % and 20 %. To prepare sample film for the dielectric property test, 1:1 molar ratio of epoxy (YD-128, Kukdo Chemical Co., Korea) and the three synthesized SMA copolymers were mixed with addition of 2-imidazole as a hardening enhancer. The film was baked on aluminum foil from 70 oC to 150 oC for four hours. The film thicknesses were kept as 0.8 mm.”
For the reaction of SMA and epoxy, the epoxy was a commercialized one, not synthesized by authors. Therefore, it was very hard to provide the detailed experimental data with the commercialized one. The composition of the epoxy was not open to authors, and it block to show or provide explainable data without knowing detailed components.
(6) Why there is no structure and properties investigation for the sample containing epoxy?
Response: Authors would like to thank reviewer for helpful comments. Authors included more discussions upon the sample containing epoxy as follows:
In lines of 336-339,
“After preparations of prepreg films with the epoxy resin, measured Tgs were obtained as increased to 165.9 oC for the unmodified SMA, 173.2 oC for the acrylate modified SMA, and 172.8 oC for the bismaleimide modified SMA, respectively.”
(7) What is the Mn, Mw and PDI for the different obtained polymer? Which is very important for the synthetic polymer.
Response: Authors would like to thank reviewer for helpful comments. Authors provide additional table of Table 1 for the measured Mn, Mw, PDI, and Tg in the revised manuscript. It was provided in Table 1 and detailed experimental method was given, separately.
In lines of 139-147:
“Characterization of physical properties of the unmodified SMA, acrylic SMA, bismaleimide SMA copolymers right after their syntheses were performed to measure number averaged molecular weight (Mn), weight averaged molecular weight (Mw), polydispersity index (PDI), and glass transition temperature (Tg), respectively. The Mn, Mw, and PDI were measured by gel permeation chromatography (GPC; 1260 Infinity II, Agilent Co., USA). The Tg was measured by differential scanning calorimeter (DSC; 131evo, Setaram KEP Technologies, Switzerland) under N2 environment with programmable temperature gradient of 20 oC/min.
Chemical and electrical characterizations of the unmodified SMA, acrylic SMA, bismaleimide SMA copolymers have been compared respectively.”
(8) How is the thermal and mechanical properties for the obtained polymers?
Response: Authors would like to thank reviewer for helpful comments. Authors provide additional table of Table 1 having the thermal properties of Tg for the unmodified, acrylate modified, and bismaleimide modified SMAs without combination of epoxy.
Reviewer 2 Report
I thank the authors for a useful manuscript, which deals with a very interesting and current issue. The issue of printed circuit boards for high-frequency applications is important and I appreciate the authors' attention to this issue. Permittivity and dissipation factor can vary significantly depending on frequency but also depending on operating temperature. A material that is suitable for use at kHz may no longer be suitable for applications requiring frequencies on the order of MHz or even GHz. The introduced material is certainly promising, but I just find the described sample preparation for testing very abstract and would appreciate some schematics or photographs of sample fabrication for testing and shaping for analysis.
The article lacked for me at least brief information about the mechanical properties of the evaluated SMA materials. The article is not primarily focused on this, but if the material has the potential to replace epoxy resin in conventional PCB substrates, I would be interested to know if this is realistic in terms of mechanical properties as well.
In the case of the article in the prestigious journal, the ten used literature sources seem to me as a small number. I do not require the addition of specific other literature sources, rather I would just give some more thought to adding some sources with a slight extension of the article in relation to, e.g., more detailed information on the frequency and temperature dependencies of the permittivity and dissipation factor of the epoxy resin-based reference material.
Below I present comments, which I divide into major and minor comments. The major comments are related to the scientific content of the manuscript and need to be incorporated. Minor comments are more related to the formalities of the manuscript.
Major comments
On the line 70 you state that the epoxy resin shows high dielectric losses "high dielectric loss (Df : > 0.1)" - do you have these values of dissipation factor verified by your own measurements or are you referring to the literature? If you are using the literature, there should be a citation after the value. Not only frequency but also temperature has an important role to influence the dissipation factor - do dissipation factor values greater than 0.1 apply over the entire range of standard operating temperatures?
In the case of information on dielectric measurements, it would be useful to add the temperature at which the measurements were performed, the measurement voltage and information on the geometry of used electrode system.
The data from the NMR analyses in Figure 3 are almost identical at first glance in the given view a, b and c, and if a certain peak shift is to be visible from the plots, it would be useful to show the details of the curves for a certain narrow interval in ppm. For the less experienced reader, the assignment of numbered carbons to the peaks of the NMR record is not clear. I recommend a better description of Figure 3 in the context of the text below the figure. The caption of Table 1 states "chemical shift of Figure 3." - Rather, it is the chemical shift (shown) in Figure 3.
The resolution of Figure 4 is set to a different resolution than Figure 3 - I would ask for a unification of this and a justification as to why the intensity on the y-axis is only quantified in Figure 4.
The results of the dielectric analyses are shown as box plots in Figure 5, but there is nothing in the text about performing more than one measurement for a one material. Please add information in the measurement definition about the number of measurements and the performance of the basic statistical analysis.
Minor comments
The notation of the dielectric constant (permittivity) is not uniform, but currently in the English literature I most often see the notation ε (or εr). It is to be considered whether this notation is better than the Dk notation.
For the dissipation factor, tgδ is more commonly used, but Df is also acceptable.
The formulation on lines 16 and 17 "Dk dielectric characteristics of..." does not seem to me to be entirely appropriate - when it is expanded it is the dielectric constant dielectric characteristics of... I would rather just state dielectric constant characteristics.
On line 33 you give the formulation "low dielectric (low k or low Dk) materials..." which is not clear. I would rather state Materials with low dielectric constant / with low dielectric losses / with low dissipation factor. It is not defined what the k in brackets means.
On line 34 and 35 you explain the operation delay (τ) twice - you only need to explain the abbreviation once with first usage in the text.
On lines 41 and 42 you give the formulation "and binder to versatile applications including paper and microelectronics" - I understand the meaning, but I would better separate paper and microelectronics in that sentence.
Why do you use a tilde instead of a hyphen in the list of multiple literature sources? I would list [1-4] by default. Similarly, I would use a hyphen in the interval of temperatures, etc.
I would not have stated the assignment of donors and acceptors to styrene and maleimide as "electron donor (ST) and acceptor (MA)" - this is evoking the definition of abbreviations.
Edit "simply synthesized simply" on line 50 - just once simply.
Instead of " leaky current" use leakage current and instead of "energy loss" I would use energy losses.
On line 85, I would not use "the signal delay, τ," but just the signal delay or τ - explained above.
On line 90 you repeatedly explain the abbreviation NMR.
On line 100 you list "into" twice - please check for duplication.
I'm not a native English speaker, but I wouldn't start a sentence with "To measure / To analyze", but I would ideally start with a subject.
Please unify the definition of manufacturer and type of all used instruments.
On lines 133 and 135 you mention Figure 1 once and then Fig. 1 - please unify the format throughout the document.
In calculations (1) to (4) and in the subsequent Table 3, I would list "mol %" as the unit, i.e., in brackets after the identification of the material component.
The y-axis in Figure 5b does not show "dielectric loss" but dissipation factor.
Author Response
Reply and Response (Polymers 2281088)
*All corrected paragraphs in revised manuscript were blue-colored.
Corrected parts were highlighted in blue color, at the same time, “track changes” function was in mode during revision.
Reviewer #2 :
I thank the authors for a useful manuscript, which deals with a very interesting and current issue. The issue of printed circuit boards for high-frequency applications is important and I appreciate the authors' attention to this issue. Permittivity and dissipation factor can vary significantly depending on frequency but also depending on operating temperature. A material that is suitable for use at kHz may no longer be suitable for applications requiring frequencies on the order of MHz or even GHz. The introduced material is certainly promising, but I just find the described sample preparation for testing very abstract and would appreciate some schematics or photographs of sample fabrication for testing and shaping for analysis.
Response: Authors would like to thank reviewer for encouraging comments. Also, the prepared powder sample photographs were added in Figure 1 and test fixture of dielectric measurement having epoxy-mixed prepreg film was given in Figure 5(b).
The article lacked for me at least brief information about the mechanical properties of the evaluated SMA materials. The article is not primarily focused on this, but if the material has the potential to replace epoxy resin in conventional PCB substrates, I would be interested to know if this is realistic in terms of mechanical properties as well.
Response: Authors would like to thank reviewer for helpful comments. Authors would like to thank reviewer for helpful comments. Authors provide additional table of Table 1 for the measured Mn, Mw, PDI, and Tg in the revised manuscript.
In lines of 165-170:
“Table 1 shows number average molecular weight (Mn), averaged molecular weight (Mw) and polydispersity index (PDI) and Tg of the unmodified SMA, acrylic SMA, and bismaleimide SMA copolymers, respectively. With the modification of acrylate and bismaleimide to the SMA, both Mn and Mw increased as expected, which could provide suitable thermal stability with the higher Tg for the high frequency applications. In addition, the bismaleimide modified SMA shows the lowest values of PDI.”
In lines of 280-288:
“As shown in Table 1, polymerization properties of the acrylate or bismaleimide modified SMA copolymers exhibited higher Tg than the unmodified SMA copolymer, which would induce thermal stability upon PCB application under ultrahigh frequency. In fact, to enforce the thermal stability, inorganic nanocomposites have been continuously introduced [5]. However, there could be a well mixing or blending difficulties with the inorganic composites [4,5]. In addition, various initiator concentrations, ST:MA feed ratios (1:1, 2:1, 4:1, 6:1), and reaction temperatures (110 oC, 115 oC, 120 oC, 125 oC) have been examined in this study. In Table 1, the reaction conditions achieving PDI value under 4.0 were kept for all SMA copolymers [19].”
In the lines of 135-138:
“For dielectric properties, Agilent N5230A microwave network analyzer (1 MHz to 10 GHz) was used to directly measure er and Df values of the SMA and modified SMA polymer’s flat film (0.8 mm) at room temperature without any support and any electrode. For the measurement, five different spots were tested and averaged values were plotted.”
In the case of the article in the prestigious journal, the ten used literature sources seem to me as a small number. I do not require the addition of specific other literature sources, rather I would just give some more thought to adding some sources with a slight extension of the article in relation to, e.g., more detailed information on the frequency and temperature dependencies of the permittivity and dissipation factor of the epoxy resin-based reference material.
Response: Authors would like to thank reviewer for helpful comments. The journal office requested 30 reference papers. So, authors added more reference papers to cover more general issues with styrene, SMA, epoxy, and low k dielectric layer. There are four epoxy resin-based reference papers in revised manuscript. In addition, it is not easy to change temperature for the prepreg film at test fixture. So, authors provided more details about the measurement.
Test fixture of dielectric measurement having epoxy-mixed prepreg film was given in Figure 5(b).
Below I present comments, which I divide into major and minor comments. The major comments are related to the scientific content of the manuscript and need to be incorporated. Minor comments are more related to the formalities of the manuscript.
Major comments
On the line 70 you state that the epoxy resin shows high dielectric losses "high dielectric loss (Df : > 0.1)" - do you have these values of dissipation factor verified by your own measurements or are you referring to the literature? If you are using the literature, there should be a citation after the value. Not only frequency but also temperature has an important role to influence the dissipation factor - do dissipation factor values greater than 0.1 apply over the entire range of standard operating temperatures?
Response: Thank for an important point. The value was from an input from PCB industry in Korea. Therefore, authors removed the number.
In the case of information on dielectric measurements, it would be useful to add the temperature at which the measurements were performed, the measurement voltage and information on the geometry of used electrode system.
Response: Authors would like to thank reviewer for helpful comments. Here, there is no need to have voltage to measure the dielectric characterization with Agilent N5230A Microwave Network Analyzer (1 MHz to 10 GHz). That is why authors included a photograph in Fig. 5(b). The measurement was done at room temperature. So, authors corrected Materials and Methods part in lines of 135-138:
“For dielectric properties, Agilent N5230A microwave network analyzer (1 MHz to 10 GHz) was used to directly measure er and Df values of the SMA and modified SMA polymer’s flat film (0.8 mm) at room temperature without any support and any electrode. For the measurement, five different spots were tested and averaged values were plotted.”
The data from the NMR analyses in Figure 3 are almost identical at first glance in the given view a, b and c, and if a certain peak shift is to be visible from the plots, it would be useful to show the details of the curves for a certain narrow interval in ppm. For the less experienced reader, the assignment of numbered carbons to the peaks of the NMR record is not clear. I recommend a better description of Figure 3 in the context of the text below the figure. The caption of Table 1 states "chemical shift of Figure 3." - Rather, it is the chemical shift (shown) in Figure 3.
Response: Authors would like to thank reviewer for helpful comments. The Figure 3 was replotted based on magnification about the major peaks. In addition, wrong information in Table 3 were corrected in blue color in the revised manuscript.
The resolution of Figure 4 is set to a different resolution than Figure 3 - I would ask for a unification of this and a justification as to why the intensity on the y-axis is only quantified in Figure 4.
Response: Authors would like to thank reviewer for helpful comments. The resolution was increased with Figure 4. In Figure 4, the peak intensities were different among three SMA copolymers. It was a reason to remain the y-axis label.
The results of the dielectric analyses are shown as box plots in Figure 5, but there is nothing in the text about performing more than one measurement for a one material. Please add information in the measurement definition about the number of measurements and the performance of the basic statistical analysis.
Response: Authors would like to thank reviewer for helpful comments. Authors put a proper discussion which can be informative to readers in revised manuscript.
So, authors corrected Materials and Methods part in lines of 135-138:
“For dielectric properties, Agilent N5230A microwave network analyzer (1 MHz to 10 GHz) was used to directly measure er and Df values of the SMA and modified SMA polymer’s flat film (0.8 mm) at room temperature without any support and any electrode. For the measurement, five different spots were tested and averaged values were plotted.”
Minor comments
The notation of the dielectric constant (permittivity) is not uniform, but currently in the English literature, I most often see the notation ε (or εr). It is to be considered whether this notation is better than the Dk notation.
Response: Authors would like to thank reviewer for helpful comments. Author exchanged the Dk notation to εr.
For the dissipation factor, tgδ is more commonly used, but Df is also acceptable.
Response: Authors would like to thank reviewer for helpful comments. Since the Df is acceptable, it was remained without change.
The formulation on lines 16 and 17 "Dk dielectric characteristics of..." does not seem to me to be entirely appropriate - when it is expanded it is the dielectric constant dielectric characteristics of... I would rather just state dielectric constant characteristics.
Response: Authors would like to thank reviewer for helpful comments. Authors corrected the statement and toned down the significance of stereochemistry and crystallization.
In lines of 18-21:
“The low er dielectric characteristics of the modified SMA copolymer could be correlated with effects from stereo-structure of carbon chains or conformational orientation, where degree of crystallization was analyzed by X-ray diffraction (XRD) and nuclear magnetic resonance (NMR) spectroscopies.”
On line 33 you give the formulation "low dielectric (low k or low Dk) materials..." which is not clear. I would rather state Materials with low dielectric constant / with low dielectric losses / with low dissipation factor. It is not defined what the k in brackets means.
Response: Authors would like to thank reviewer for helpful comments. Authors removed the expression of “k” in brackets.
In lines of 18-21:
“The low er dielectric characteristics of the modified SMA copolymer could be correlated with effects from stereo-structure of carbon chains or conformational orientation, where degree of crystallization was analyzed by X-ray diffraction (XRD) and nuclear magnetic resonance (NMR) spectroscopies.”
On line 34 and 35 you explain the operation delay (τ) twice - you only need to explain the abbreviation once with first usage in the text.
Response: Authors would like to thank reviewer for helpful comments. The doubly written tau was removed.
On lines 41 and 42 you give the formulation "and binder to versatile applications including paper and microelectronics" - I understand the meaning, but I would better separate paper and microelectronics in that sentence.
Response: Authors would like to thank reviewer for helpful comments. The introduction part was more specified with paper and microelectronic parts.
In lines of 45-47:
“The paper application may contain non-sticky coating on paper surface [11,16]. The microelectronic applications have been resorted to PCBs formations in most cases [12-15].”
Why do you use a tilde instead of a hyphen in the list of multiple literature sources? I would list [1-4] by default. Similarly, I would use a hyphen in the interval of temperatures, etc.
Response: Authors would like to thank reviewer for helpful comments. The tildes were substituted with “hypen”.
I would not have stated the assignment of donors and acceptors to styrene and maleimide as "electron donor (ST) and acceptor (MA)" - this is evoking the definition of abbreviations.
Response: Authors would like to thank reviewer for helpful comments. The concept or definition of donor and acceptor was removed.
In lines of 52-55:
“Spontaneous arrangement between nonpolar styrene and polar maleic anhydride into alternating sequences has been widely studied as a result of electron donors with the ST and acceptor with the MA complex formation at low temperature condition (60 oC~ 80 oC) [14].”
Edit "simply synthesized simply" on line 50 - just once simply.
Response: Authors would like to thank reviewer for helpful comments. The typo was removed.
Instead of " leaky current" use leakage current and instead of "energy loss" I would use energy losses.
Response: Authors would like to thank reviewer for helpful comments. The better descriptions suggested by reviewer were taken.
In lines of 79-84:
“Frequently, a high er means poor insulating property having a leakage current [25, 26]. In addition, a high Df refers to the energy losses that could be changed into internal layer in the forms of vibrational or thermal energy, when the voltage is applied to the nonconductive material. To diminish the energy losses from the sole epoxy material, other resins could have been mixed together for a long time due to improved mechanical properties and compatibility advantage even with glass fibers.”
On line 85, I would not use "the signal delay, τ," but just the signal delay or τ - explained above.
Response: Authors would like to thank reviewer for helpful comments. It was corrected as “tau” t.
On line 90 you repeatedly explain the abbreviation NMR.
Response: Authors would like to thank reviewer. The repeated one was removed.
On line 100 you list "into" twice - please check for duplication.
Response: Authors would like to thank reviewer. The repeated one was removed.
I'm not a native English speaker, but I wouldn't start a sentence with "To measure / To analyze", but I would ideally start with a subject.
Response: Authors would like to thank reviewer for kind comments. Such expressions like “To measure/To analyze” were removed.
Please unify the definition of manufacturer and type of all used instruments.
Response: Authors would like to thank reviewer for kind comments. The expression for the used instrument was unified.
On lines 133 and 135 you mention Figure 1 once and then Fig. 1 - please unify the format throughout the document.
Response: Authors would like to thank reviewer. The format was unified.
In calculations (1) to (4) and in the subsequent Table 3, I would list "mol %" as the unit, i.e., in brackets after the identification of the material component.
Response: Authors would like to thank reviewer for helpful comments. All mol % units were changed to mole fraction.
The y-axis in Figure 5b does not show "dielectric loss" but dissipation factor.
Response: Authors would like to thank reviewer. The notation in Fig. 5(b) was changed into “dissipation factor”.
Round 2
Reviewer 1 Report
All of the issues mentioned were resolved in detail
Author Response
Reviewer #1 :
All of the issues mentioned were resolved in detail
Response: Authors would like to thank reviewer for evaluation.

Reviewer 2 Report
I thank to the authors for responding to all my comments and consider the performed edits of the article as sufficient. I accept the article and I see it as a significant improvement compared to the first version.
I would only ask for the meaning of the abbreviation DI - to be added or replaced by the full term on line 120 (I assume it is demineralised water, but the abbreviation is not explained). On line 144, write Tg with subscript g. In Table 1, I would unify the number of decimal points for Tg. For example, on line 329 you list the interval of values with a hyphen, then on line 331 you list the interval of values with a tilde. Sometimes you list spaces before and after these symbols - if the meaning is same, please unify throughout the document. The notation Tgs on line 339 does not seem appropriate to me and I would rather choose e.g. values of Tg.
Author Response
Reviewer #2 :
I thank to the authors for responding to all my comments and consider the performed edits of the article as sufficient. I accept the article and I see it as a significant improvement compared to the first version.
Response: Authors would like to thank reviewer for kind comments.
I would only ask for the meaning of the abbreviation DI - to be added or replaced by the full term on line 120 (I assume it is demineralised water, but the abbreviation is not explained). On line 144, write Tg with subscript g. In Table 1, I would unify the number of decimal points for Tg. For example, on line 329 you list the interval of values with a hyphen, then on line 331 you list the interval of values with a tilde. Sometimes you list spaces before and after these symbols - if the meaning is same, please unify throughout the document. The notation Tgs on line 339 does not seem appropriate to me and I would rather choose e.g. values of Tg.
Response: Authors would like to thank reviewer for helpful comments. All the commented parts were corrected in 2nd revised version.
